# Basal-Plane Catalytic Activity of Layered Metallic Transition Metal Ditellurides for the Hydrogen Evolution Reaction

**Hagyeong Kwon [1,†], Dongyeon Bae [1,†], Hyeyoung Jun [1,†], Byungdo Ji [2], Dongyeun Won [2] [image_id], Jun-Ho Lee [3] [image_id], Young-Woo Son [3], Heejun Yang [2,*] and Suyeon Cho [1,*]**

1   Division of Chemical Engineering and Materials Science, Ewha Womans University, Seoul 03760, Korea; khk960621@gmail.com (H.K.); ehddus0904@gmail.com (D.B.); jhy7636@naver.com (H.J.)
2   Department of Energy Science, Sungkyunkwan University, Suwon 16419, Korea; owenba@naver.com (B.J.); dongyeun7@gmail.com (D.W.)
3   Korea Institute for Advanced Study, Seoul 02455, Korea; junholee@lbl.gov (J.-H.L.); hand@kias.re.kr (Y.-W.S.)
*   Correspondence: h.yang@skku.edu; (H.Y.); s.cho@ewha.ac.kr; (S.C.)
†   These authors contributed equally to this work.

**Abstract:** We report the electrochemical hydrogen evolution reaction (HER) of two-dimensional metallic transition metal dichalcogenides (TMDs). $TM$Te$_2$ ($TM$: Mo, W, and V) single crystals were synthesized and characterized by optical microscopy, X-ray diffraction, and electrochemical measurements. We found that $TM$Te$_2$ acts as a HER-active catalyst due to the inherent catalytic activity of its basal planes. Among the three metallic $TM$Te$_2$, VTe$_2$ shows the best HER performance with an overpotential of 441 mV and a Tafel slope of 70 mV/dec. It is 668 mV and 137 mV/dec for MoTe$_2$ and 692 mV and 169 mV/dec for WTe$_2$. Even though VTe$_2$ has the lowest values in the exchange current density, the active site density, and turn-over-frequency (TOF) among the three $TM$Te$_2$, the lowest charge transfer resistance (R$_{CT}$) of VTe$_2$ seems to be critical to achieving the best HER performance. First-principles calculations revealed that the basal-plane-active HER performance of metallic TMDs can be further enhanced with some Te vacancies. Our study paves the way to further study of the inherent catalytic activity of metallic 2D materials for active hydrogen production.

**Keywords:** Hydrogen evolution reaction (HER); Transition metal dichalcogenide (TMD); Two-dimensional (2D) material

## 1. Introduction

Hydrogen energy is receiving tremendous attention as it is regarded as the most promising eco-friendly and renewable energy to replace the carbon emission energies such as fossil fuels [1–4]. To produce high purity hydrogen gas, electrochemical water splitting has been frequently used with noble metal-based electrocatalysts [5]. Much research has focused on the development of new types of the electrocatalysts, combining lower amounts of noble metals with earth-abundant materials for economy and efficient hydrogen production [6,7].

Hybrid electrocatalysts commonly consist of catalytic active nanoparticles embedded in catalytic inactive low-dimensional supports such as a graphene matrix with defects [8–14]. However, the high surface energy of nanoparticles has hindered the goal of obtaining small-sized nanoparticles and has boosted the aggregation of nanoparticles, which is known as the Ostwald ripening effect in nanoparticles [15,16]. Therefore, for a high number of active sites, the periodic arrangement of the catalytic active sites in the 2D plane can be a challenging theme for an efficient hydrogen evolution reaction (HER).

Two dimensional transition metal dichalcogenides (TMDs) are thought of as next generation electrocatalysts for the HER because of their abundant HER active sites with almost zero Gibbs free energy for hydrogen adsorption [17,18]. The semiconducting TMDs such as $MoS_2$ and $WS_2$ demonstrate excellent performance for hydrogen production at their active sites, mostly defects or edges [19–21]. Some TMDs experience a structural phase transition to a metallic state with defects, strains, or doping [22,23] and the thickness of TMDs has been known as an important property for the charge transfer process in HER performance [24]. Therefore, an unstable surface state due to many defects or edges is the main obstacle for stable hydrogen production.

The recently reported basal plane activity in metallic TMDs has been spotlighted to solve the above problems [18,23–26]. Soek et al. reported that the nonhomogeneous distribution of surface atoms creates intrinsic catalytic activity at its basal plane [22]. They also reported that the HER process of metallic TMDs sometimes accompanies structural modification due to hydrogen adsorption, which helps to enhance the catalytic activity of metallic TMDs. Kwon et al. also reported that the HER performance of metallic TMDs can be enhanced by the creation of Te vacancies [27]. Even though many metallic TMDs show the HER performance, the detailed mechanism of their basal plane-catalytic activity for HER is still unclear.

Here, we report the intrinsic catalytic activity of the basal plane of metallic transition metal (*TM*) ditellurides (*TM*Te$_2$), where the *TM*s are V, Mo, and W. We synthesized *TM*Te$_2$ single crystals using previously reported methods [27–29] and characterized it by optical microscopy, X-ray diffraction, and electrochemical measurements. X-ray diffraction results confirmed that *TM*Te$_2$ single crystals have the orthorhombic, monoclinic, and monoclinic structure for WTe$_2$, MoTe$_2$, and VTe$_2$, respectively [27–29]. The optical microscopy and the SEM images (Figure S1) show that their surfaces were well prepared with the clean surface minimizing the cracks, the edges, or the defects, which were usually regarded as extrinsic active sites in many TMDs. We found that clean surfaces of *TM*Te$_2$ behaved as HER-active surfaces due to the inherent catalytic activity of their basal planes. Among the three metallic *TM*Te$_2$, VTe$_2$ achieved the best HER performance with the lowest overpotential and Tafel slope, which demonstrated that it was promising for active hydrogen production.

## 2. Experiments

X-ray diffraction of three *TM*Te$_2$ single crystals was taken using an X-ray diffractometer (D/MAX-2500/PC, Rigaku, Japan) with Cu K radiation ($\lambda$ = 1.54718 Å) at room temperature. The temperature dependent resistance of three *TM*Te$_2$ single crystals was measured by a physical property measurement system (PPMS) (Quantum Design) through direct current (DC) measurements over a temperature range between 1.8 and 300 K.

For the electrochemical measurement of three *TM*Te$_2$ single crystals, a piece of *TM*Te$_2$ single crystal was glued to an Au (100 nm)/SiO$_2$ (300 nm)/Si substrate and connected to the bottom Cu plate. Except for the open area for the HER, all other surface areas were covered by an insulating nail polish to prevent any unexpected leakage current. The open area of samples for the HER was automatically estimated by the Leopard 2.5 program installed in the software for the optical microscope; the current density (*J*) in this manuscript was calculated by the current divided by the open sample area.

All electrochemical experiments were performed using a potentiostat (VMP3 from Bio Logic, France) and three-electrode electrochemical cell in 0.5 M sulfuric acid (H$_2$SO$_4$) electrolyte solution. A silver/silver chloride electrode (Ag/AgCl, Qrins, Korea), a platinum wire electrode (Qrins, Korea), and the *TM*Te$_2$ were used as the reference, counter, and working electrodes, respectively. Linear sweep voltammetry (LSV) was conducted with a scan rate of 5 mV/s. The electrochemical active surface area (ECSA) analysis with VTe$_2$, MoTe$_2$, and WTe$_2$ was conducted at different voltage range and sweep rates. For impedance measurement, potentiostatic electrochemical impedance spectroscopy (PEIS) technique was used and measurements were carried out at frequency range between 100 kHz to 100 mHz with an AC voltage of 10 mV and a DC voltage, electric potential, of 390 mV (vs. RHE). The results were analyzed by using z-fitting software named EC-Lab.

First-principles calculations were conducted using a Quantum ESPRESSO package [30]. We used norm-conserving pseudopotentials [31] for W, Te, and H atoms, a plane wave energy cutoff of 80 Ryd, and a Perdew-Burke-Ernzerhof exchange-correlation functional [32]. Equilibrium lattice parameters of a = 6.36 Å and b = 3.53 Å are used to simulate the orthorhombic MoTe$_2$ structure.

## 3. Result and Discussion

The atomic arrangements of the three metallic transition metal ditellurides (*TM*Te$_2$), VTe$_2$, WTe$_2$, and MoTe$_2$, are shown in Figure 1a. One transition metal atom is surrounded by six tellurium atoms with the form of a distorted octahedron: one center *TM* atom and six corner Te atoms. These layered metallic TMDs have exotic buckled layer structures due to the lattice distortion originating from strong correlations between spin, charge, and lattice in 2D materials [28,29]. The buckled monolayer of *TM*Te$_2$ has non-uniform distribution of surface Te atoms, which is categorized as α (orange circles) and β (yellow circles) according to the vertical positions at the basal plane as seen in Figure 1a. Different from the zig-zag chains (black lines) of Mo or W atoms in MoTe$_2$ and WTe$_2$, VTe$_2$ has mirrored double zig-zag chains (black lines) of V atoms, leading to the different surface densities of site α and β for hydrogen evolution as described in Table 1.

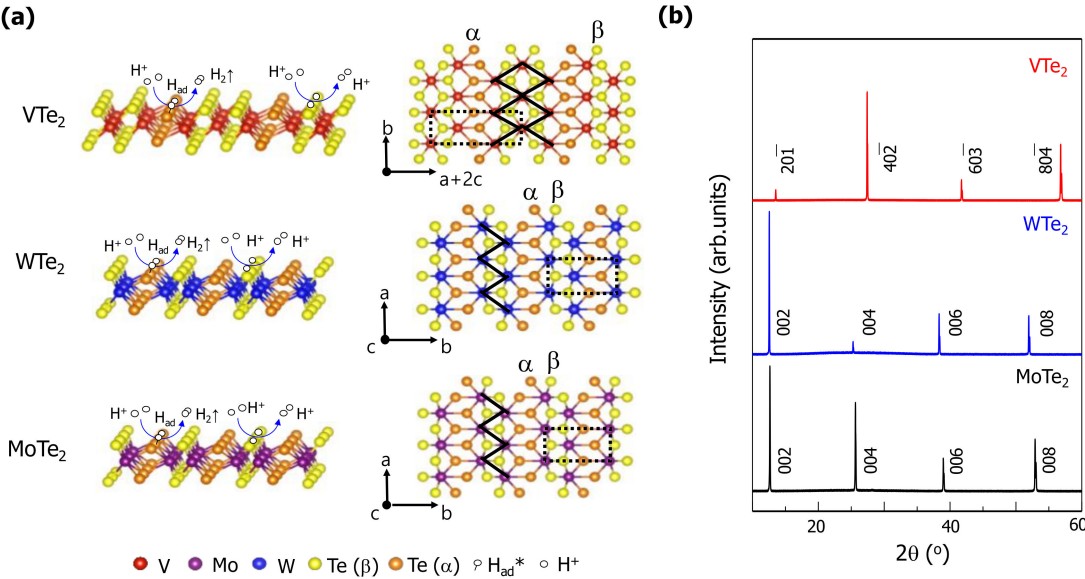

**Figure 1.** (**a**) Schematics of monolayer *TM*Te$_2$ (TM = V, W, and Mo) with side and top view. Protons (H+) in acidic electrolyte (0.5 M H$_2$SO$_4$) are just passing without reactions at catalytic inactive Te atoms (β site, yellow circles), but are adsorbed at catalytic active Te atoms (α site, orange circles) and subsequently are converted to hydrogen gas (H$_2$). (WTe$_2$; a = 3.498 Å and b = 6.338 Å, MoTe$_2$; a = 3.487 Å and b = 6.366 Å, VTe$_2$; b = 3.605 Å, a+2c = 9.529 Å). (**b**) X-ray diffraction of three *TM*Te$_2$ single crystals.

Different Te atoms in the buckled layer of the three *TM*Te$_2$ have different Gibbs free energies for hydrogen adsorption. We classified two Te atomic sites as α sites and β sites to analyze the HER performances of metallic *TM*Te$_2$. We assume that α site Te atoms are better catalytically active than β site Te atoms in the three *TM*Te$_2$, which is consistent with recently reported HER works on MoTe$_2$ and WTe$_2$ [22,23,27]. During the HER process, approaching proton atoms (H$^+$) are captured at α sites rather than β sites due to the differences in Gibbs free energies, which creates the intrinsic catalytic activity for hydrogen evolution [22,23,27]. The number densities of active α site in three *TM*Te$_2$ are calculated with unit surface area, which contains one active α site (squares with dashed lines in Figure 1a) in the basal plane (see Table 1).

**Table 1.** Summary of electrochemical properties of three *TM*Te$_2$.

| Catalyst | Overpotential (mV vs. RHE) | Tafel Slope (mV/dec) | Exchange Current Density (mA/cm$^2$) | * site density (#/nm$^2$) | TOF (ms$-$) | Double Layer Capacitance (mF/cm$^2$) | Charge Transfer Resistance (k*$_{CT}$) |
|---|---|---|---|---|---|---|---|
| VTe$_2$ | −441 | 70 | 0.2 | 2.92 | 2.14 | 0.03 | 5.8 |
| MoTe$_2$ | −668 | 137 | 0.7 | 4.50 | 4.85 | 0.08 | 13.0 |
| WTe$_2$ | −692 | 169 | 1.0 | 4.51 | 6.92 | 0.06 | 21.6 |

To study the inherent catalytic activity of the basal plane in *TM*Te$_2$, *TM*Te$_2$ single crystals were used for this study. We synthesized single crystalline *TM*Te$_2$ using the NaCl-Flux method and the solid state reaction described in our previous studies [28,29]. We confirmed that *TM*Te$_2$ single crystals have the orthorhombic (a = 3.498 Å, b = 6.338 Å, c = 15.432 Å, β = 90°), monoclinic (a = b = 3.559 Å, c = 40.693 Å, β = 90°), and monoclinic (a = 14.826 Å, b = 3.605 Å, c= 9.366 Å, β = 112.4°) structure for WTe$_2$, MoTe$_2$, and VTe$_2$, respectively, and the *TM*Te$_2$ was well grown with a growth axis in the (00n) direction using x-ray diffraction as seen in Figure 1b. All three *TM*Te$_2$ show metallic behaviors in which the resistivity decreases with a decreasing temperature (see Figure S2). The three *TM*Te$_2$ are known as semimetals (or bad metals) with few carriers due to the quasi gap at the Fermi surface [33,34]. Thus, they have good charge transfer and also less of a screening effect.

Figure 2a shows the experimental setup for electrochemical measurements of the three metallic *TM*Te$_2$. We used a three-electrode system for the TMTe$_2$ working electrode (WE), platinum wire for the counter electrode (CE), and Ag/AgCl for the reference electrode (RE). We applied an external voltage between the WE and RE and measured the current density between the WE and CE. *TM*Te$_2$ single crystals were used for the WE and the sample area of the WE was estimated by an analyzing program (Leopard 2.5) with optical images.

The electrochemical HER performance of the three *TM*Te$_2$ electrodes was examined by linear sweep voltammetry (LSV) and electrochemical impedance spectroscopy (EIS). Figure 2b shows that the LSV curves of three *TM*Te$_2$ electrodes and the overpotentials of −441, −668, and −692 mV were required to obtain 10 mA/cm$^2$ for VTe$_2$, MoTe$_2$, and WTe$_2$, respectively. The overpotential of VTe$_2$ was found to be the smallest among three *TM*Te$_2$ electrodes. Figure 2c shows the Tafel slopes of the three *TM*Te$_2$ electrodes were 70, 137, and 169 mV/dec for VTe$_2$, MoTe$_2$, and WTe$_2$, respectively, and VTe$_2$ exhibited the lowest Tafel slope among them. The overpotential and the Tafel slope of VTe$_2$ single crystal were consistent with reported values in micro-sized VTe$_2$ flakes, which were transferred to the glassy carbon working electrode by drop-cast method, indicating that the basal plane of VTe$_2$ can be active for HER [35]. We extracted the exchange current densities of three *TM*Te$_2$ from extended linear lines of Tafel slopes down to zero value in the overpotential, 0.2, 0.7, and 1.0 μA/cm$^2$ for VTe$_2$, MoTe$_2$, and WTe$_2$, respectively.

To analyze the catalytic activity of the α site in the *TM*Te$_2$, we calculated the turn over frequency (TOF), which is proportional to the conversion rate from proton to hydrogen gas at one active site for one second. The TOF was calculated by equation $TOF = \frac{j \times A \times N_A}{S \times n \times F}$, where $j$ is the exchange current density, $n$ is the number of electrons transferred per molecule (2 for HER), $F$ is the Faraday constant (96,458 C mol$^{-1}$ electrons), $A$ is the superficial electrode area, and $N_A$ is Avogadro's number ($6.022 \times 10^{23}$ mol$^{-1}$). The TOF values were 2.14, 4.85, and 6.92 ms$^{-1}$ for VTe$_2$, MoTe$_2$, and WTe$_2$, respectively, and WTe$_2$ had the largest TOF value among the three. It seems that the TOF results contradict the reported Gibbs free energies of α sites in MoTe$_2$ ($\Delta G_H$ = 0.77 eV) and WTe$_2$ ($\Delta G_H$ = 1.14 eV) [17,18]. This is explained by the fact that the hydrogen adsorption at the catalytic active sites is less critical in determining the hydrogen conversion rate during the HER process.

To study the charge transfer process for the HER, we measured the electrochemical impedance spectroscopy (EIS) with the three *TM*Te$_2$ electrodes. Figure 2d shows the Nyquist plots of the three *TM*Te$_2$ electrodes with the complex impedance, the real part Z$_{Re}$, and the imaginary part −Z$_{Im}$ with various frequencies from 100 kHz to 100 mHz. We included the Warburg impedance (w) on the

equivalent Randles circuit models to interpret the mass transfer at the low frequency region (inset in Figure 2d). The Nyquist plots show semicircles in the kinetic control region at the high frequencies with the diameter of $R_{CT}$, which represents the charge transfer resistance. The values of $R_{CT}$ were 5.8, 13.0, and 21.6 k$\Omega$, which increased in order of VTe$_2$, MoTe$_2$, and WTe$_2$ and the lowest $R_{CT}$ was found in VTe$_2$.

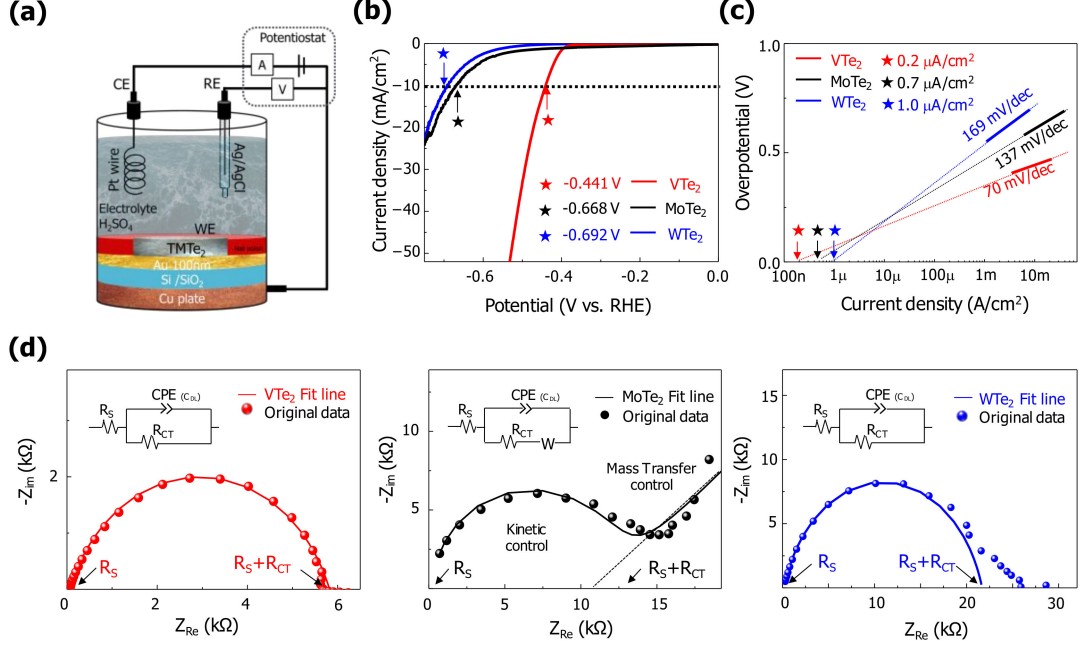

**Figure 2.** (**a**) Experimental set-up for electrochemical measurements of three *TM*Te$_2$ single crystals. (**b**) Linear sweep voltammetry and (**c**) Tafel plots and (**d**) the electrochemical impedance spectroscopy of three *TM*Te$_2$ with the Randles circuit, which include the series resistance ($R_S$), the charge transfer resistance ($R_{CT}$), and the constant phase element (CPE).

We also estimated the double layer capacitance ($C_{DL}$) between the *TM*Te$_2$ electrode and the electrolyte. Cyclic voltammetry (CV) was measured with different scan rates. Figure 3a shows the hysteresis of the CV curves taken in non-Faradaic voltage regions without any additional electrochemical reactions. We plotted the curves of the capacitive current density as shown in the CV curves in Figure 3a as a function of scan rate to obtain $C_{DL}$ of three *TM*Te$_2$ electrodes. Figure 3b shows similar $C_{DL}$ values due to the similar sample area, the same dielectric constant of the electrolyte, and the same distance of the double layer capacitance. These $C_{DL}$ values in *TM*Te$_2$ single crystals were almost 100 times smaller than the reported $C_{DL}$ in micro-sized flakes on the glassy carbon working electrode due to the difference of the electrochemical surface area [23].

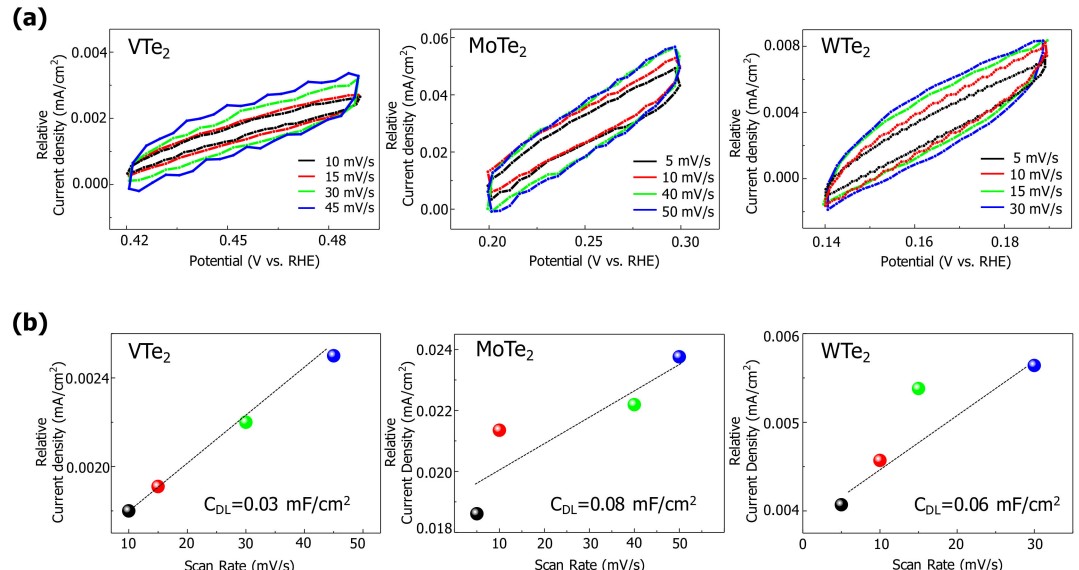

**Figure 3.** (**a**) Cyclic voltammetry (CV) curves at different scan rates and (**b**) capacitive current as a function of scan rate of VTe$_2$, MoTe$_2$, and WTe$_2$.

We summarized our experimental results of the HER performances observed in the three *TM*Te$_2$ in Table 1. Among the three *TM*Te$_2$, VTe$_2$ had the best HER performance with a small overpotential and Tafel slope. Even though VTe$_2$ had the smallest α site density and TOF, the smallest R$_{CT}$ was found in VTe$_2$. It seems that the HER process of *TM*Te$_2$ is significantly influenced by the charge transfer process. Therefore, to enhance the inherent catalytic activity of *TM*Te$_2$, we can further enhance the charge transfer for a small R$_{CT}$ or to increase the electrochemical active surface area for large C$_{DL}$.

Considering the Gibbs free energies of α sites in *TM*Te$_2$, the defect creation can help to enhance the hydrogen adsorption on basal plane of *TM*Te$_2$. Figure 4 shows the energies of hydrogen adsorption of α and β sites in MoTe$_2$ with or without Te defects. We can decrease the energy of hydrogen adsorption at α sites from 0.84 eV to 0.21 eV with Te defects, which can be a considerable value for better HER.

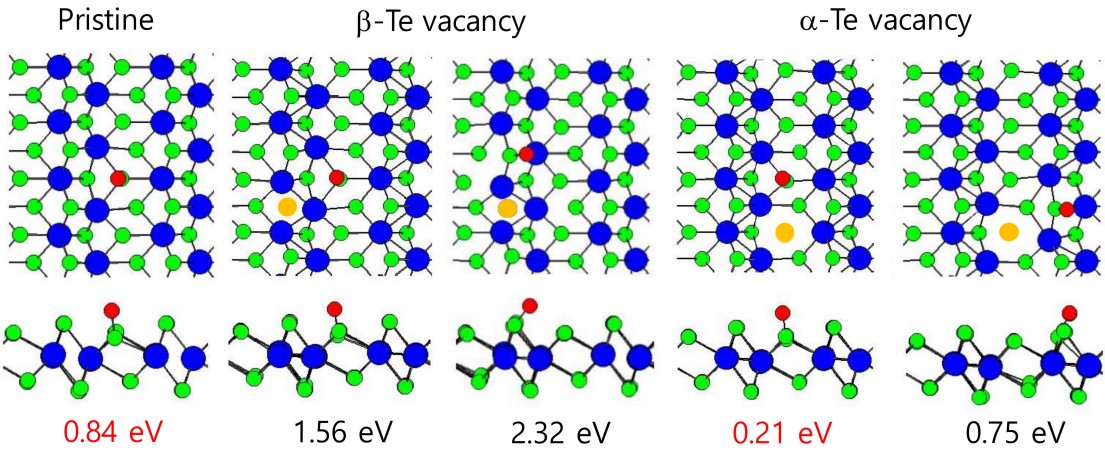

**Figure 4.** Hydrogen adsorption energies at α and β sites of *TM*Te$_2$ with or without Te vacancies.

## 4. Conclusions

In conclusion, the basal plane catalytic activity for the HER was investigated with three metallic *TM*Te$_2$ (*TM*: Mo, W, and V). We measured the electrochemical measurements, LSVs, EISs, and CVs, and obtained the overpotential, Tafel slope, exchange current, α site density, TOF, C$_{DL}$, and R$_{CT}$. Among the three *TM*Te$_2$, VTe$_2$ had the best HER performance with an overpotential of −441 mV and

a Tafel slope of 70 mV/dec. Even though $VTe_2$ had the lowest exchange current, the $\alpha$ site density and TOF among the three $TMTe_2$, the lowest $R_{CT}$ of $VTe_2$ seems to be critical to achieve the best HER performance. Our study suggests that the inherent HER performance of metallic TMDs can be enhanced by enhancing the charge transfer process, increasing the electrochemical active surface area or by creating Te vacancies.

**Supplementary Materials:** The following are available online at http://www.mdpi.com/2076-3417/10/9/3087/s1, Figure S1: Optical image and SEM image of $WTe_2$ single crystal for the electrochemical measurements, Figure S2: Residual resistance ratio (RRR) of three $TMTe_2$ single crystals.

**Author Contributions:** H.K.: Data curation, Investigation, Writing —Original draft, Writing—review & editing. D.B.: Data curation, Investigation, Writing—Original draft. H.J.: Data curation, Investigation. B.J.: Investigation, D.W.: Investigation. J.-H.L.: Formal analysis. Y.-W.S.: Formal analysis. H.Y.: Conceptualization, Supervision. S.C.: Conceptualization, Supervision, Writing—Original draft, Writing - review & editing. All authors have read and agreed to the published version of the manuscript.

**Acknowledgments:** S.C.: H.K. and D.B. were supported by the Basic Science Research Program through the National Research Foundation of Korea (NRF) funded by the Ministry of Science, ICT, and Future Planning (2020R1A2C2003377). H.K. and H. J. were supported by the National Research Foundation of Korea (NRF) and Center for Women In Science, Engineering and Technology (WISET) grant funded by the Ministry of Science and ICT (MSIT) under the team research program for female engineering students. H.Y. acknowledges support from the National Research Foundation of Korea (NRF) under Grant No. NRF-2020R1A2B5B02002548. Y.-W.S. was supported by the NRF of Korea (Grant No. 2017R1A5A1014862, SRC program: vdWMRC Center). We thank the Korea Institute for Advanced Study for providing computing resources (KIAS Center for Advanced Computation Linux Cluster System) for this work.

**Conflicts of Interest:** The authors declare no competing financial interests.

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
