# Peer review of "Basal-Plane Catalytic Activity of Layered Metallic Transition Metal Ditellurides for the Hydrogen Evolution Reaction"

_applsci, doi:10.3390/app10093087_

Round 1
Reviewer 1 Report
Report attached.

Author Response
We upload our response as a Word file.
Please see the attachment.

Reviewer 2 Report
Authors present and interesting work about the basal plane catalytic activity for the HER in metallic MoTe2, WTe2 and VTe2, with detailed electrochemical experiments and theoretical calculations. The manuscript is well written but more detailed explanations regarding some points, would be advisable. Moreover, topological characterization of the defect-free crystals is required.
Specifically:
-In the introduction, it is always interesting to cite reviews about the field, better than specific papers, or at least both of them. I will recommend authors to include some reviews. For example, ref. 5 it is too specific (moreover, I could not find the full text in English), it could also include recent reviews like “Recent Advances in Noble Metal (Pt, Ru, and Ir)-Based Electrocatalysts for Efficient Hydrogen Evolution Reaction. ACS Omega 2020, 5, 1, 31-40” or a different one, but more general.
-Line 64: “We confirmed that the single crystalline TMTe2 surfaces were well prepared without any cracks, edges or defects, which were usually regarded as extrinsic active sites in many TMDs”.
This is a very important claim and there is not a single proof of it in the manuscript. Which is the size of the crystal in the electrochemical cell? How can be possible that there are no cracks or edges in a large crystal? How is the topological characterization performed? The abstract talks about optical measurements, but they are not shown.
-Atomic arrangement of TMTe2:
- a, b, c axes should be showed in figure 1a.
- For a better understanding for all readers, discussion about TMTe2 phases should be included. Comparing with ref 22, X-Ray diffractogram VTe2 is CDW phase, which are MoSe2 and WSe2, are they 1T?.
- Regarding fig. 1a line 116 claims “The number densities of active a site in three TMTe2 are calculated with unit surface area, which contains one active a site (squares with dashed lines in Figure 1a), in the basal plane” a sites are orange circles, so I see 1 + ½+ ½ = 2 inside each rectangle with dashes lines in all three TMTe2. What am I doing wrong?. Moreover, to estimate nm2 it is necessary to show the length of the rectangle with dashes lines.
-Figure 1c: Legend reads: “Residual resistance ratio (RRR) of three TMTe2 single crystals.” Are the values in the inset RRR=r(300K)/ r (0K)? Could this be better specified? From my point of view, the legend describes r(T)/ r (300K) as RRR and it is confusing.
-line 131: Counter electrode Glassy Carbon? Are authors sure about this? In the experimental section the counter electrode is described as Pt wire (line 78), that is more common counter electrode.
-Figure 2a: What is the red layer where the TMD is placed on?
-Line 159: Figure 2d instead of 2c?
-Figure 3b: Which equation is used to estimate the double layer capacitance? Are y axes in a and b actually the same?
-Why First-principles calculations conducted using a Quantum ESPRESSO package are only performed for WTe2? Which could be the results expected for VTe2 and MoTe2?
-It would be interesting that authors compare the obtained values (overpotential, Tafel slope, exchange current…) with previously reported values in literature and comment about them.

Author Response

(The authors gave the same response as above.)

Round 2
Reviewer 2 Report
I have add my comments along the document, between authors answers. I am afraid I still see several unclear points.

Round 3
Reviewer 2 Report
These are my last comments about the previous manuscript (applsci-761589) entitled "Basal-Plane Catalytic Activity of Layered Metallic Transition Metal Ditellurides for Hydrogen Evolution Reaction" by H. Kwon et al. submitted to Applied Science,
-Line 59 should add SEM in the list.
-I do not know if there is a problem in my pdf viewer but there is nothing in the SEM image (only a grey rectangle).
-My point about why the authors did not use VTe2 as test case is actually not answered. I understand that their last point is that the Gibb energy is not the important thing, but this does not explain for me why they do not carry out their theoretical calculations on their most interesting material (VTe2).
Author Response
These are my last comments about the previous manuscript (applsci-761589) entitled "Basal-Plane Catalytic Activity of Layered Metallic Transition Metal Ditellurides for Hydrogen Evolution Reaction" by H. Kwon et al. submitted to Applied Science,
Thanks for the comments, we have responsed the reviewers's the comments like below,
-Line 59 should add SEM in the list.
Actually, we do not understand what the reviewer said. We already added SEM images in figure S1 and revised the sentence (line 59) in the previous rebuttal letter. If the reviewer has unsatisfied things, please tell us more clearly.
-I do not know if there is a problem in my pdf viewer but there is nothing in the SEM image (only a grey rectangle).
We show that SEM image to say there are no cracks and no edges at the sample surface. There are nothing at the sample surface. So, there are nothing in SEM image.
-My point about why the authors did not use VTe2 as test case is actually not answered. I understand that their last point is that the Gibb energy is not the important thing, but this does not explain for me why they do not carry out their theoretical calculations on their most interesting material (VTe2).
Our study claimed that HER performance of metallic TMDs can be determined by the charge transfer process rather than Gibbs free energy. We do not want to put the Gibbs free energy of all samples because that seems to be not a critical parameter for HER of metallic TMDs.